# Can Large Language Models Help Experimental Design for Causal Discovery?

## Abstract

Designing proper experiments and intervening targets is a longstanding problem in scientific or causal discovery. It is fundamentally impossible to identify the underlying causal structure merely based on the observational data. Obtaining interventional data, on the other hand, is crucial to causal discovery, yet it is usually expensive or time-consuming to obtain sufficient interventional data to facilitate causal discovery. Previous approaches usually leverage uncertainty or gradient signals to determine the intervention targets, and may suffer from the suboptimality. In this work, we investigate a different approach, whether we can leverage Large Language Models (LLMs) to assist with the intervention targeting in causal discovery by making use of the rich world knowledge about the experimental design in LLM. Specifically, we present **L**arge Languag**e** Model **G**uided **I**ntervention **T**argeting (LeGIT), a robust framework that effectively incorporates LLMs to assist with the intervention targeting in causal discovery. Surprisingly, across 4 different scales of realistic benchmarks, LeGIT significantly outperforms previous approaches. LeGIT opens up a new frontier for using LLMs in experimental design.

## 1 Introduction

Science originates along with discovering new causal knowledge with *interventional experiments inspired by observations* (Hanson, 1958; Kuhn & Hawkins, 1963). The art of finding causal relations from different interventions is then summarized and improved with statistical methods (Pearl & Mackenzie, 2018; Spirtes et al., 2000; 2010; Glymour et al., 2019). Identifying and utilizing causal relations is essential to a variety of applications such as biology (Vowels et al., 2022) and financial system (Dong et al., 2023). Despite the wide deployment of causal discovery methods, identifying the underlying causal connections merely based on observational data is fundamentally impossible (Spirtes et al., 2000). It usually requires additional interventional data obtained by perturbing part of the causal system to overcome the limited identifiability issue (Spirtes et al., 2000).

Nevertheless, collecting interventional data is expensive and time-consuming, as it usually involves a physical process of a real-world system (Cherry & Daley, 2012; Sunar et al., 2019). Consequently, *both the number of samples and intervention targets are significantly limited in real-world experimental design* (Murphy, 2006; Tong & Koller, 2001). Previous approaches usually leverage uncertainty (Lindley, 1956) or information theoretic metric to maximize the utility of an experiment (Tigas et al., 2022; Zhang et al., 2022). Recently, leveraging gradient signals for intervention targeting has gained significant success (Olko et al., 2023), as it naturally fits into various gradient-based causal discovery methods (Lippe et al., 2022b). Despite some success, both uncertainty-based and gradient-based approaches may still suffer from suboptimality, as the estimation of the signals is usually noisy (Olko et al., 2023) and can easily mislead the intervention targeting.

The emergence of large language models (LLMs) (Brown et al., 2020; OpenAI, 2022; Ouyang et al., 2022; Touvron et al., 2023; OpenAI, 2023; Bubeck et al., 2023), provides an opportunity to incorporate world knowledge about experimental design into the intervention targeting process. It therefore raises an intriguing research question:

> *Can we incorporate the knowledge of LLMs to assist with intervention targeting?*

In fact, early explorations with LLMs in multiple causal learning and reasoning tasks show that LLMs may have already captured a large amount of domain knowledge (Kiciman et al., 2023; Lampinen

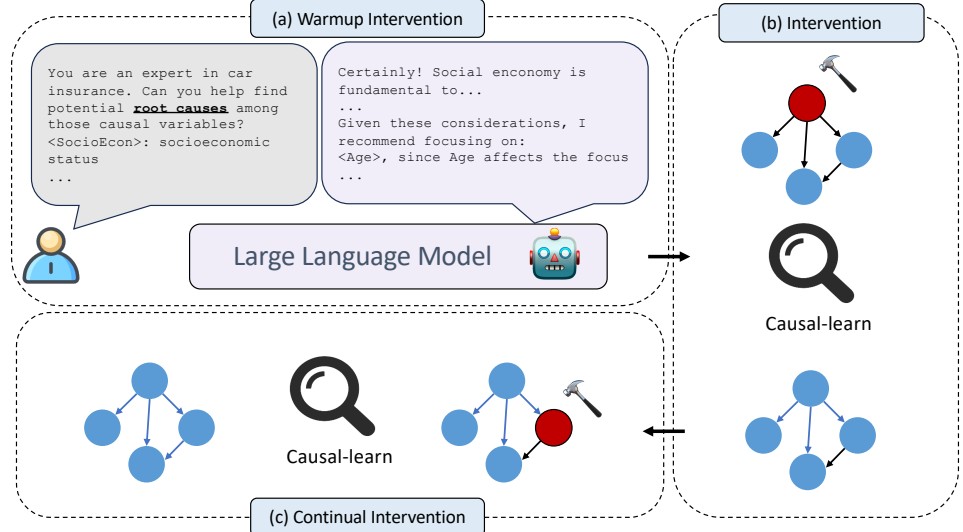

Figure 1: Illustration of LeGIT framework. At the beginning, in step (a), LLMs are mainly used to warm up the causal discovery process. Leveraging the rich world knowledge and relating it to the meta information of the task, in step (b), LLMs can help identify influential nodes that facilitate the discovery of the skeleton of the causal graph. Through multiple rounds of steps (a) and (b), we can identify relatively clearer causal structures, such that previous methods can provide high-quality guidance to continue finding informative intervention targets in step (c).

et al., 2023; Abdulaal et al., 2024). LLMs are shown to be able to process the meta-information encoded in natural language, and leverage the meta-information to reason for the causality, which was considered to be restricted to humans (Gopnik et al., 2004; Trott et al., 2022; Sahu et al., 2022). In addition, discussions about the limitations of LLMs in understanding causality were also raised in the community (Zečević et al., 2023; Jin et al., 2023a;b; Zhang et al., 2023a). Therefore, it requires a robust trade-off that maximally extracts the world knowledge in LLMs about the experimental design, while not being misled by the hallucinations of LLMs about the causality (Zhang et al., 2023c).

To this end, we present a new framework called **L**arge Languag**e** Model **G**uided **I**ntervention **T**argeting (LeGIT) that aims to maximally while robustly leveraging the knowledge in LLMs to assist with the intervention targeting. Shown as in Fig. 1, at the beginning of the causal discovery, the numerical-based methods have limited numerical knowledge about the underlying causal system to use. Consequently, the estimated signals tend to be noisy and misleading. In contrast, LLMs can leverage the meta-information about the causal system and relate the learned world knowledge to identify high-potential intervening targets. After obtaining a relatively clearer causal graph, LLMs may not be able to provide sufficient guidance. Therefore, similar to humans, LeGIT leverages numerical methods to select the intervening targets. Our contributions can be summarized as follows:

- To the best of our knowledge, we are the first to investigate the use of LLMs into the experimental design to select intervention targets for causal discovery.
- We propose a novel framework called LeGIT that combines the advantages of both the previous numerical methods as well as the LLMs to facilitate the intervening targetting.
- We conduct extensive experiments with 4 real-world benchmarks and verify that LeGIT can empower numerical-based methods and achieve state-of-the-art performance.

## 2 RELATED WORK

**Intervention/Experiment Design** Scientific progress in causal discovery is often driven by inter-ventional experiments inspired by observational insights (Hanson, 1958; Kuhn & Hawkins, 1963). Traditional methods focused on designing effective experiments to establish causal links, while statis-tical approaches aimed to automate causal inference from observational data (Pearl & Mackenzie,

2018; Spirtes et al., 2000; 2010; Vowels et al., 2022). However, observational data alone is insufficient for identifying causal structures, and interventional data is costly to collect (Spirtes et al., 2000). To address these challenges, several methods for optimal intervention design have been developed.

Active Intervention Targeting (AIT) selects intervention targets using an $F$-test inspired criterion, evaluating discrepancies in interventional sample distributions from a posterior distribution of graphs (Scherrer et al., 2021). Causal Bayesian Experimental Design (CBED) uses Bayesian Optimal Experimental Design to select interventions that maximize mutual information (MI) between new data and existing graph beliefs, with MI estimated via a BALD-like method (Tigas et al., 2022; Houlsby et al., 2011). Gradient-based Intervention Targeting (GIT) Olko et al., 2023 leverages gradient information to determine interventions that maximize impact on causal parameter updates, which is particularly advantageous in low-data settings. Causal Active Learning for Optimal Intervention Design Zhang et al., 2023b takes an active learning approach, using Bayesian updates to iteratively choose interventions that most effectively reduce uncertainty in achieving a target outcome. In our work, we explore leveraging these advanced intervention strategies within the framework of LLMs to determine whether LLMs can effectively engage in experimental design for causal discovery, pushing the boundaries of what automated, data-driven causal inference can achieve.

**Causal Discovery With LLMs** Recent advancements in large language models (LLMs) like ChatGPT (OpenAI, 2022) have opened new opportunities in causal inference by incorporating domain knowledge, common sense, and contextual reasoning into the causal discovery process (Kiciman et al., 2023). LLMs have demonstrated capabilities across Pearl's ladder of causation—association, intervention, and counterfactuals—bridging gaps that traditional models have with high-level causal reasoning. They have shown promising results in pairwise causal discovery tasks by utilizing semantic information not accessible through numerical data alone (Kiciman et al., 2023).

Despite these advances, challenges remain. LLMs can sometimes behave like "causal parrots", repeating learned associations without demonstrating true causal reasoning (Zečević et al., 2023). Moreover, their performance varies significantly depending on task complexity, with limited success in advanced causal reasoning such as full graph discovery and counterfactual analysis (Zhang et al., 2023a; Jin et al., 2023b;a; Long et al., 2023c). Another promising line of work integrates LLMs with traditional causal discovery methods to leverage their complementary strengths (Long et al., 2023a; Abdulaal et al., 2024; Liu et al., 2024). This hybrid approach has shown improved performance in constructing causal graphs, benefiting from LLMs' understanding of language context and traditional methods' data-driven precision.

While these studies highlight the use of LLMs in causal analysis, the question of whether LLMs can effectively contribute to experimental design in causal discovery remains largely unexplored. Designing experiments involves proposing interventions, predicting outcomes, and evaluating experimental strategies—tasks that require more than mere causal inference. This paper aims to fill this gap by exploring the potential of LLMs to assist in experimental design, evaluating their strengths and limitations in guiding causal experiments.

## 3 PRELIMINARIES

This work focuses on leveraging LLMs to select proper intervention targets in an online causal discovery setting (Lippe et al., 2022b; Olko et al., 2023). We begin by briefly introducing the preliminaries and notations of this work.

### 3.1 CAUSAL STRUCTURE DISCOVERY

The causal relations between different variables can be formulated using the structural causal models (SCM) (Pearl & Mackenzie, 2018; Spirtes et al., 2000; 2010; Glymour et al., 2019; Vowels et al., 2022). More specifically, in an SCM, we are given $n$ endogenous variables $X = (X_1, ..., X_n)$, where the generation process of each variable can be expressed as $X_i = f_i(PA_i, U_i)$ where $PA_i$ is the set of variables that are the causal parents of $X_i$, and $U_i$ is the external independent noise when generating $X_i$.

The causal relations between $n$ variables can be further characterized via a direct acyclic graph (DAG), $G = (V, E)$, where $V = \{1, ..., n\}$ is the set nodes corresponding to the set of random variables $\{X_1, ..., X_n\}$. Each edge $(i, j) \in E$ in the edge set $E$ refers to the relation of direct cause $X_i \in PA_j$, i.e., $X_i$ is one of the causes of the variable $X_j$. The joint distribution of all the variables associated with the DAG can be expressed as $P(X_1, ..., X_n) = \Pi_{i=1}^n P(X_i|PA_i)$.

---

**Algorithm 1** ONLINE CAUSAL DISCOVERY (Olko et al., 2023)

**input** causal discovery algorithm $\mathcal{A}$ (e.g., ENCO,), intervention targeting method $\mathcal{M}$, number of data acquisition rounds $T$, observational dataset $\mathcal{D}_{obs}$

**output** final parameters of graph model: $\varphi_T$ and CausalDAG: $\mathbb{P}(G)$

1: $\mathcal{D}_{int} \leftarrow \varnothing$
2: Fit graph model $\varphi_0$ with algorithm $\mathcal{A}$ on $\mathcal{D}_{obs}$
3: **for** round $i = 1, 2, \ldots, T$ **do**
4:     $I \leftarrow$ generate intervention targets using $\mathcal{M}$
5:     $\mathcal{D}_{int}^I \leftarrow$ query for data from interventions $I$
6:     $\mathcal{D}_{int} \leftarrow \mathcal{D}_{int} \cup \mathcal{D}_{int}^I$
7:     Fit $\varphi_i$ with algorithm $\mathcal{A}$ on $\mathcal{D}_{int}$ and $\mathcal{D}_{obs}$
8: **end for**

---

Causal structure discovery aims to identify the underlying DAG $G$. However, when given only the joint observed distribution $P(X_1, ..., X_n)$, it does not uniquely determine a DAG, as there might be different DAGs that can generate the same joint distribution. On the contrary, the observational data can merely determine a set of DAGs up to a Markov Equivalence Class (MEC) (Spirtes et al., 2000).

## 3.2 ONLINE CAUSAL DISCOVERY

To identify the underlying ground truth DAG from the MEC, interventional data is widely incorporated into the causal discovery process (Tong & Koller, 2001; Hauser & Bühlmann, 2011; Ke et al., 2019). Hence, online causal discovery is proposed to overcome the issue (Ke et al., 2019; Olko et al., 2023).

As given in Algorithm 1, an online causal discovery procedure is built upon a causal discovery algorithm $\mathcal{A}$ that is able to leverage both the observational data and interventional data to recover the underlying causal structure. More formally, the interventional data is usually obtained through single-node intervention on some causal variable $X_i$. The intervention will replace the generation process of $X_i$ with a new distribution, for which we denote as $\widehat{P}(X_i|PA_i)$ (Pearl & Mackenzie, 2018). Then, it yields an interventional distribution:

$$P_i(X) = \widehat{P}(X_i|PA_i)\Pi_{j\neq i}P(X_j|PA_j), \tag{1}$$

where the node $i$ is called the intervention target. An intervention can be both hard and soft. A hard intervention directly removes the dependency of $X_i$, i.e., $\widehat{P}(X_i|PA_i) = \widehat{P}(X_i)$; Otherwsie soft.

The online discovery will proceed by $T$ rounds. At the beginning of the first round, an initial graph model $\phi_0$ is fitted based on the observational data. Then, in the follow-up $T$ rounds, an intervention target $I$ will be selected using some intervention targeting method. For each selected $I$, a batch of samples will be obtained and be integrated into all interventional data to execute the causal discovery algorithm $\mathcal{A}$. After $T$ rounds, the fitted DAG will be the final output.

Previous approaches may use different intervention targeting methods. For example, Scherrer et al. (2021) propose Active Intervention Targeting (AIT) to select the desired intervention targets based on the $F$-test. Tigas et al. (2022) approximate the posterior distribution over all possible DAGs and leverage Bayesian Optimal Experimental Design to select the most informative intervention targets.

Different from the Bayesian approaches, Olko et al. (2023) propose Gradient-based Intervention Targeting (GIT), which leverages the gradient signals from the gradient-based causal discovery methods to estimate the utility of each intervention target via hallucinated gradients (Ash et al., 2020). Due to the natural combination of the gradient-based causal discovery methods and the GIT method, GIT achieves significant performance improvements over previous Bayesian-based approaches. Therefore, in this work, our follow-up discussion will center on the gradient-based approaches, i.e., the GIT method and the ENCO causal discovery methods(Lippe et al., 2022b).

# 4 LARGE LANGUAGE MODEL GUIDED INTERVENTION TARGETING

Despite the success of GIT method, similar to other estimation-based approaches, GIT is highly sensitive to the accuracy of the gradient estimation. Therefore, the existence of noises in the estimated scores can easily mislead the intervention targeting.

## 4.1 CHALLENGES IN EXISTING INTERVENTION TARGETING

To demonstrate the aforementioned issue and the challenges in the existing intervention targeting methods more concretely, we consider three realistic causal discovery benchmarks, i.e., alarm (Beinlich et al., 1989), child (Dempster, 1993) and insurance (Binder et al., 1997) and plot the score distribution for the intervention targeting.

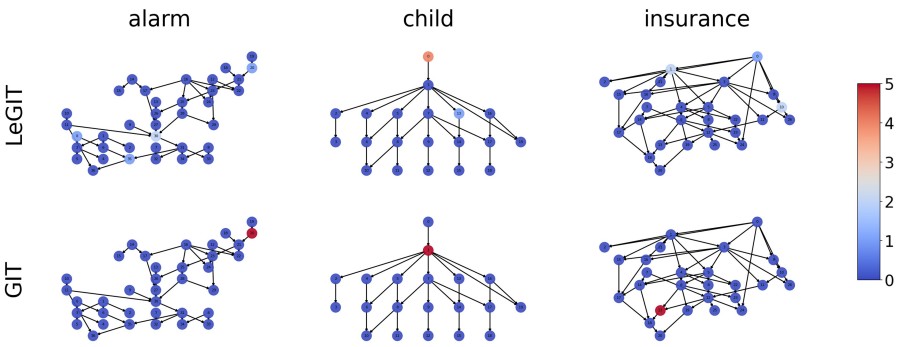

Figure 2: Initial intervention targeting of LLM-based selection and the gradient-based selection.

As given in Fig. 2, it can be found that the success of GIT varies across different datasets. Intuitively, at the beginning of the intervention, intervening on variables that affect a lot of other variables can bring more information about the system (Lindley, 1956). However, from the distribution of the GIT estimated scores, it can be found that most of the variables share similar scores. In the benchmark of child, the selected intervention target is an influential node. However, in insurance, the selected node simply influences few nodes. Intervening on the selected target with limited influence may cause significant resource waste, and further misleads the followup online causal discovery rounds.

In contrast, we construct prompts to inquiry LLMs about the root causes in this system, given only the meta information such as simple variable descriptions. The specific prompts are given in Fig. 3, and the suggested intervening targets are also highlighted in Fig. 2. It can be found that, given only the meta information, LLMs are able to relate the rich world knowledge to locate the desired influential nodes.

```
You are a helpful assistant and expert in alarm
system research. Assuming we can do interventions
to all the variables, your job is to assist in
designing the best intervention experiments among
the following variables to help discover their
causal relations:

<variable name>: variable description

Assuming we can do interventions to all the
variables, given the aforementioned variables and
their descriptions, can you echo your knowledge
about those variables, temporally analyze their
relations, and then choose the best 5
intervention targets from all the variables which
hopefully are the root causes of the other
variables to start our analysis of their causal
relations?

Let's think and analyze step by step. Then,
provide your final answer (variable names only)
within the tags <Answer>...</Answer>, separated
by ", ".
```

Figure 3: Prompt template at the initial round.

## 4.2 LARGE LANGUAGE MODEL GUIDED INTERVENTION TARGETING

Motivated by the aforementioned experiments, we present our framework **Large Language** Model **Guided Intervention Targeting** (LeGIT) to combine the strengths of both numerical-based methods and LLMs to facilitate the intervention targeting. The algorithm description of LeGIT is given in Algorithm 2. LeGIT consist of three stages.

---

**Algorithm 2** LeGIT: LARGE LANGUAGE MODEL GUIDED INTERVENTION TARGETING

---

**input** causal discovery algorithm for Intervention Data $\mathcal{A}$ (e.g., ENCO); Intervention Score targeting method $\mathcal{M}_I$ (e.g GIT); LLM for root cause proposal $\Psi$, number of data acquisition rounds $T$; Observational dataset $\mathcal{D}_{obs}$; Graph Node List $V$; Warmup Epoch $T_{warmup}$; Missing Search Epoch $T_{missing}$

**output** final parameters of graph model: $\varphi_T$ and CausalDAG: $\mathbb{P}(G)$

1: //Get Warmup List from LLM
   $\mathcal{D}_{warmup} \leftarrow \Psi(V, T_{warmup})$
2: **for** round $i = 1, 2, \ldots, T$ **do**
3:     **if** i $<= T_{warmup}$ **then**
4:         $D_{int}^I \leftarrow \mathcal{D}_{warmup}[i]$
5:     **else if** i $= T_{warmup} + 1$ **then**
6:         // Get the Isolated (Missing) Nodes List
           $V_{missing} \leftarrow$ isolated node from $\mathbb{P}(G_i)$
7:         //Get Intervention Target from Missing Node List
           $\mathcal{D}_{missing} \leftarrow \Psi(V_{missing}, T_{missing})$
8:         $D_{int}^I \leftarrow \mathcal{D}_{missing}[i - T_{warmup}]$
9:     **else if** $T_{warmup} < i <= T_{warmup} + T_{missing}$ **then**
10:        $D_{int}^I \leftarrow \mathcal{D}_{missing}[i - T_{warmup}]$
11:    **else if** $T_{warmup} + T_{missing} < i <= 2(T_{warmup} + T_{missing})$ **then**
12:        //Double Selection LLM'S List
           $D_{int}^I \leftarrow (D_{warmup} + D_{missing})[i - T_{warmup} - T_{missing}]$
13:    **else**
14:        $D_{int}^I \leftarrow$ generate intervention targets using $\mathcal{M}_I$
15:    **end if**
16:    $\mathcal{D}_{int} \leftarrow \mathcal{D}_{int} \cup \mathcal{D}_{int}^I$
17:    Fit $\varphi_i$ with algorithm $\mathcal{A}$ on $\mathcal{D}_{int}$ and $\mathcal{D}_{obs}$
18: **end for**

---

**Warmup Stage** Since at the very beginning of the online causal discovery, numerical-based estimations are noisy and easily mislead the online causal discovery, we begin by prompting LLMs to relate the pre-trained knowledge, analyze the variable description, and suggest influential candidates. The prompt template is given in Fig. 3. The prompting will give the beginning list of intervention targets $\mathcal{D}_{warmup}$. From $\mathcal{D}_{warmup}$, we will select $T_{warmup}$ variables to obtain a basic map of the underlying causal system.

**Bootstrapped Warmup Stage** Although the first warmup stage yields a basic structure of the underlying causal system, due to the intrinsic limitations of LLMs such as limited context length (Liu et al., 2023) and hallucination (Zhang et al., 2023c), LLMs may only focus on a subset of the variables and find the influential nodes therein. Nevertheless, when the number of causal variables is large, LLMs tend to give an incomplete set of influential nodes. Therefore, we further incorporate a second warmup stage, to bootstrap the use of LLM's world knowledge in early intervention targeting.

More concretely, we leverage the intermediate causal discovery results $\phi_{T_{warmup}}$ after the first $T_{warmup}$ rounds, and examine the left variables that have not been involved in $\phi_{T_{warmup}}$. Then, we further prompt LLMs to give more focus on the left set of variables and to find the influential variables that were missing in previous rounds.

In addition, since we have already obtained relatively high-quality intermediate causal discovery results, we can also incorporate the numerical-based methods to suggest a set of promising candidates

for LLMs to choose. As the numerical-based methods may still not be stable given the first $T_{warmup}$ warmup rounds, we still encourage LLMs to determine the finalist.

**Continual Intervention Stage** After the two warmup stages, we have already obtained relatively clearer yet complicated causal graphs. Even for humans, it is hard to determine the best experimental design. Therefore, we switch to using the numerical-based methods to continue to consume the remaining intervention budgets.

### 4.3 THEORETICAL DISCUSSION

After setting up the LeGIT algorithm, we now briefly discuss the convergence of LeGIT. Since LeGIT ends up with a numerical-based methods to conclude the online causal discovery, intuitively, given any numerical-based methods, such as GIT (Olko et al., 2023), and a useful online causal discovery algorithm, such as ENCO (Lippe et al., 2022b), LeGIT can converge. Nevertheless, due to a better warmup strategy in LeGIT, empirically, we find that LeGIT can converge to a better solution even when compared to the same numerical-based methods without LLMs involved.

### 4.4 PRACTICAL DISCUSSION

Following the practice in the literature, we mainly adopt GIT as the numerical-based method $\mathcal{M}$, and ENCO as the gradient-based causal discovery method. Nevertheless, as also suggested in GIT (Olko et al., 2023), ENCO can also be switched to other gradient-based methods. In addition, LeGIT is also compatible with other numerical-based approaches.

## 5 EXPERIMENTS

In this section, we conduct extensive experiments to evaluate LeGIT on real-world datasets and compare LeGIT against various baselines in intervention selection. We provide a brief overview of the experimental setups here, with further details available in Appendix A.

### 5.1 EXPERIMENTAL SETUP

**Datasets** Specifically, we use four real-world benchmark datasets along with their corresponding ground truth causal graphs from the BN repository (Scutari, 2010): *Asia*, *Child*, *Insurance*, and *Alarm*. The BN repository provides causal graphs derived from real-world applications that are widely recognized as benchmark datasets. These datasets encompass a diverse set of professional scenarios, ranging from car insurance to medical systems, which are crucial for enhancing the knowledge captured by large language models (LLMs).

1. *Asia* (Lauritzen & Spiegelhalter, 2018) dataset consists of 8 variables related to a lung cancer diagnosis system, with 8 edges.
2. *Child* (Dempster, 1993) dataset contains 20 nodes and 25 edges, modeling congenital heart disease in newborns.
3. *Insurance* (Binder et al., 1997) dataset includes 27 nodes and 52 edges, representing a car insurance system.
4. *Alarm* (Beinlich et al., 1989) dataset comprises 37 nodes and 46 edges, simulating an alarm message system for patient monitoring.

**Baselines.** We compare **LeGIT** against different online causal discovery algorithm **GIT** (Olko et al., 2023), **AIT** (Scherrer et al., 2021) as active learning online intervention selection strategies, as well as three random baselines:

1. **Random Choice**: At each step, a target node is chosen uniformly at random from the set of all nodes;
2. **Round Robin**: At each step, a target node is chosen uniformly at random from the set of unvisited nodes. Once all nodes have been selected, the visitation counts are reset to zero;

3. **Degree Prob Sample**: At each step, a target node is chosen at random from the set of all nodes, with the selection probability normalized according to the out-degree of each node;

Among the baselines, **Degree Prob Sample** can be considered as an oracle to LLM that adopts the out-degree of each node in the ground truth DAG.

**Implementation**   We employ the GPT-4-0125-preview API (OpenAI, 2023; 2024) for all LLM experiments. For all experiments in this section, we use **ENCO** (Lippe et al., 2022a) as the backbone causal discovery algorithm, with detailed settings provided in the Appendix. We utilize an observational dataset of size $|\mathcal{D}_{obs}| = 5000$, with $T = 33$ rounds of low-intervention sampling, each acquiring an interventional batch of $|\mathcal{D}_{int}^I| = 32$ samples, in a total of $N = 1056$ interventional samples. For **GIT** and **AIT**, we use $|\mathcal{G}| = 50$ graphs, each with $|\mathcal{D}_{G,i}| = 128$ data samples for the Monte Carlo approximation of the score. Considering the size of the real-world graph, we use $T_{warmup} = 3, T_{missing} = 2$ in LeGIT, except for the Asia dataset except for the Asia dataset due to its smaller size. Therefore, we set $T_{warmup} = 4, T_{missing} = 1$ for Asia dataset.

**Metrics**   We report the Structural Hamming Distance (SHD) (Tsamardinos et al., 2006) as the primary evaluation metric. In simple terms, SHD represents the number of edge insertions, deletions, or reversals required to transform one graph into another, lower is better.

## 5.2 EMPIRICAL RESULTS

The results of the benchmark experiments are presented in Table 1. Our method consistently outperforms the baseline approaches across four distinct domains, as indicated by the mean Structural Hamming Distance (SHD) calculated from five seeds under a low data budget. Fig. 4 illustrates the mean SHD of these methods in relation to the number of intervention samples.

Table 1: Average SHD with standard deviation (from 5 seeds), for real-world data ($T = 33$ rounds, and the total number of intervention samples is $N = 1056$).

|  | ALARM | INSURANCE | CHILD | ASIA |
|---|---|---|---|---|
| AIT | 32.80 ± 8.42 | 24.20 ± 7.47 | 9.00 ± 3.29 | 1.80 ± 0.75 |
| RANDOM CHOICE | 34.80 ± 2.32 | 26.00 ± 3.63 | 5.40± 1.20 | 1.20 ± 0.40 |
| ROUND ROBIN | 25.00 ± 1.26 | 25.80 ± 2.93 | 3.40 ± 2.50 | 1.40 ± 0.49 |
| DEGREE PROB | 29.40 ± 4.67 | 17.40 ± 4.54 | 6.20 ± 2.48 | 1.00 ± 0.00 |
| GIT | 19.60 ± 3.77 | 16.40 ± 3.14 | 2.80 ± 0.75 | 1.00 ± 0.00 |
| **LEGIT** | **18.80 ± 1.33** | **15.80 ± 3.11** | **2.20 ± 1.30** | **0.80 ± 0.75** |

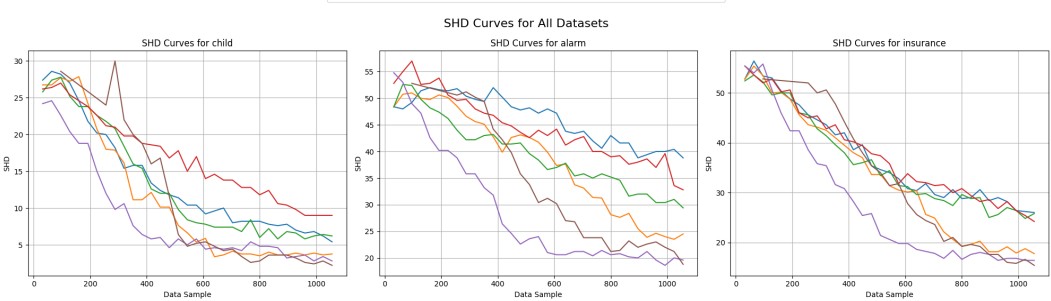

Figure 4: SHD metric for differecent methods towards different interevention sample.

As shown in Table 1, it can be found that LeGIT achieves state-of-the-art causal discovery performances, with consistent improvements against the adopted gradient-based methods. In Fig. 4, we further plot the performances of different methods along with the increase of the data samples obtained from the intervention. It can be found that, although at the beginning of the online causal

Table 2: Average SHD with standard deviation (from 5 seeds) with Low Data, for real-world data (with $T = 33$ rounds, and a total number of intervention samples $N = 528$).

| METHOD | ALARM | INSURANCE | CHILD |
|--------|-------|-----------|-------|
| GIT | $26.20 \pm 3.71$ | $22.40 \pm 3.72$ | $7.50 \pm 1.61$ |
| LEGIT | $\mathbf{26.00 \pm 5.25}$ | $\mathbf{18.80 \pm 2.04}$ | $\mathbf{6.00 \pm 2.40}$ |

discovery, LeGIT may not demonstrate outstanding SHD results. Along with more data samples combining, LeGIT converge to a better solution faster than other methods. In contrast, despite a faster decrease speed of GIT, GIT finally converges to a suboptimal solution, which verifies our discussion.

Furthermore, we also conduct additional experiments in an extremely low data setting, where we sample only 16 data samples from each intervention. The results are given in Table 2, from which we can find that, LeGIT is able to bring larger improvements. The results demonstrate the usefulness of LeGIT in real-world experimental design, where the number of interventions and the samples are both extremely low (Tong & Koller, 2001; Olko et al., 2023).

## 5.3 ABLATION STUDIES

To understand the behavior of the LLM, we begin by examining the example response shown in Fig.5 on the Insurance dataset. The LLM follows the instructions provided in Fig.3 step by step and aligns them with its background knowledge.

```
To approach this task, we'll first echo the knowledge of
the variables provided, analyze their temporal
relations, and then identify the best 5 intervention
targets.

### Echoing the Knowledge of the Variables

1. **<ThisCarCost>**: The costs associated with the
insured car. This could be influenced by factors such as
the value of the car, damages, and insurance premiums.
2. **<Mileage>**: How much mileage is on the car. Higher
mileage can indicate more use and potentially more wear
and tear.
……
22. **<DrivingSkill>**: The skill level of the driver,
influencing driving quality and accident risk.

### Temporal Analysis

Before choosing intervention targets, we should consider
the temporal order of these variables. For instance,
**<Age>**, **<SocioEcon>**, and **<SeniorTrain>** are
more foundational and could influence many other
variables like **<DrivingSkill>**, **<DrivQuality>**,
and **<RiskAversion>**. Similarly, **<MakeModel>** and
**<VehicleYear>** would precede **<CarValue>** and
**<ThisCarCost>**.

### Choosing the Best 5 Intervention Targets

Based on the temporal analysis and aiming to identify
root causes, the best intervention targets would be
those that influence multiple other variables and are
more foundational in nature.

<Answer>Age, SocioEcon, SeniorTrain, MakeModel,
VehicleYear</Answer>
```

Figure 5: Example of LLM's response on Insurance dataset.

Figs. 6 and 7 depict the selected node distribution between epochs 0-5 and 5-10, respectively. Notably, the numerical methods (GIT, AIT) tend to get trapped in the initialization phase of the Insurance dataset, consistently selecting less central nodes in the graph, often peripheral or leaf nodes. In

contrast, our model (LeGIT) identifies that SocioEcon (socioeconomic status) plays a crucial role in the insurance system, potentially influencing car choice, driving behavior, and the ability to afford certain safety features (see Appendix B).

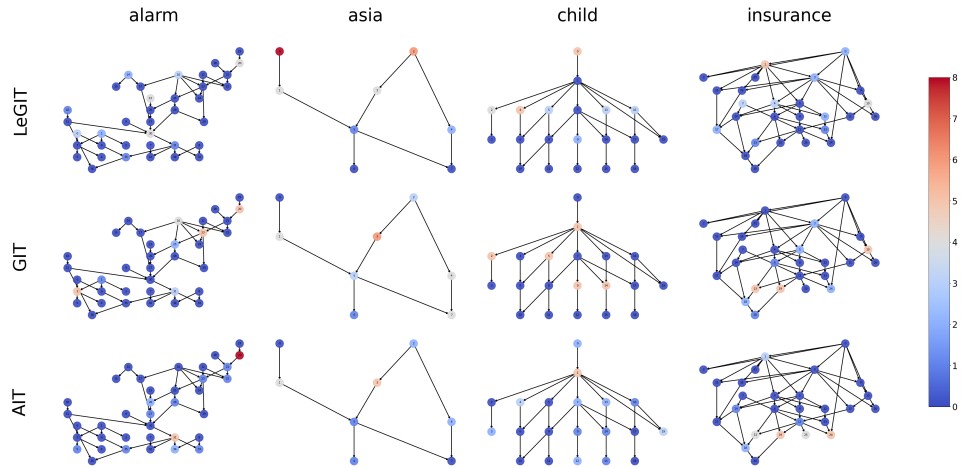

Figure 6: The selected Node Frequence obtained by different strategies on Epoch 0-5 from 5 seeds.

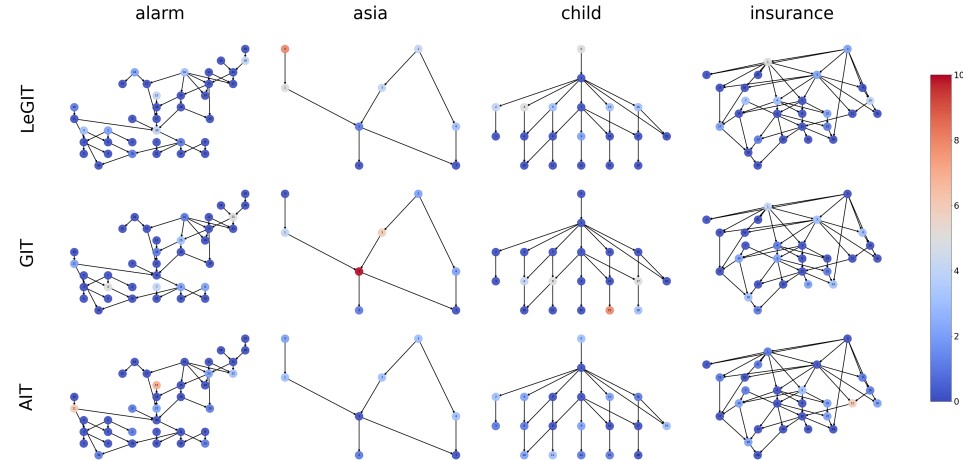

Figure 7: The selected Node Distribution obtained by different strategies on Epoch 5-10 from 5 seeds.

## 6 CONCLUSIONS

In this work, we investigated how to incorporate LLMs into the intervention targeting in experimental design for causal discovery. We introduced a novel framework called LeGIT, that combines the best of previous numerical-based approaches and the rich world knowledge in LLMs. Specifically, LeGIT leverages LLMs to warm up the online causal discovery procedure by identifying the influential root cause variables to begin the intervention. After setting up a relatively clear picture of the underlying causal graph, LeGIT then integrates the numerical-based methods to continue to select the intervention targets. Empirically, we verified the effectiveness of LeGIT that leveraging LLMs to warm up the online causal discovery can achieve the state-of-the-art performance across 4 different scale of realistic causal discovery benchmarks. Future studies can be established by a further investigation of various approaches to integrate world knowledge in LLMs for causal discovery.

ETHICS STATEMENT

This work mainly focuses on leveraging LLMs to better select the intervention targets for broader applications and social benefits. Besides, this paper does not raise any ethical concerns. This study does not involve any human subjects, practices to data set releases, potentially harmful insights, methodologies and applications, potential conflicts of interest and sponsorship, discrimination/bias/fairness concerns, privacy and security issues, legal compliance, and research integrity issues.

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

## A  MORE DETAILS OF DATASETS

In this part, we will further introduce the 4 different domain Causal graph discovery dataset from BNleaner Repository (Scutari, 2022). For the description of each variable, we refer to Long et al. (2023b) and make some changes on it.

**Asia** show as Fig.8(a) aims to model a hypothetical medical scenario in which a person visits a clinic with shortness of breath. The network helps in diagnosing the likely causes (e.g., tuberculosis, lung cancer, bronchitis) by probabilistically combining the available evidence (e.g., history of travel, smoking status, X-ray results)

**Child** show as Fig.8(b) is used to model the diagnosis of pediatric health issues, particularly those that can occur in newborns or young children. It's often employed in studies related to decision support systems, where probabilistic graphical models assist in medical diagnosis. The network is significantly larger than the Asia dataset, with 20 nodes (variables) and 25 edges.

**Insurance** shown as Fig. 8(c) intended to simulate a situation in which an insurance company needs to assess various risks and make decisions regarding policies, claims, and customer behavior. It represents the interdependencies between multiple insurance factors. It has 27 nodes and 52 edges

**Alarm** shown as Fig. 8(d) is known as the ALARM (A Logical Alarm Reduction Mechanism) network, and it was originally developed to model a patient monitoring system for anesthesia purposes. It helps in predicting physiological conditions of patients, detecting potential complications, and generating alerts when necessary, consists of 37 nodes and 46 edges.

## B  MORE DETAILS OF EXPERIMENTS

### B.1  ENCO HYPERPARAMETERS

For experiments using the ENCO framework, we used the exact parameters reported by Lippe et al. (2022a). These parameters are provided in Table 3 to ensure the completeness of our report.

Table 3: Hyperparameters used for the ENCO framework.

| parameter | value |
|---|---|
| Sparsity regularizer $\lambda_{sparse}$ | $4 \times 10^{-3}$ |
| Distribution model | 2 layers, hidden size 64, LeakyReLU($\alpha = 0.1$) |
| Batch size | 128 |
| Learning rate - model | $5 \times 10^{-3}$ |
| Weight decay - model | $1 \times 10^{-4}$ |
| Distribution fitting iterations F | 1000 |
| Graph fitting iterations G | 100 |
| Graph samples K | 100 |
| Epochs | 30 |
| Learning rate - $\gamma$ | $2 \times 10^{-2}$ |
| Learning rate - $\theta$ | $1 \times 10^{-1}$ |

### B.2  FINAL CAUSAL GRAPH

In this section, we present the final causal graph after $T = 33$, total sample $N = 1056$ results with GIT and LeGIT.

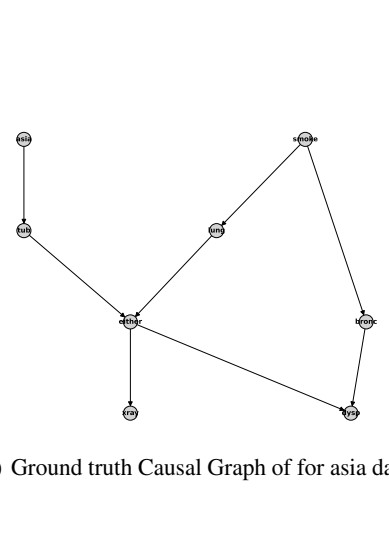
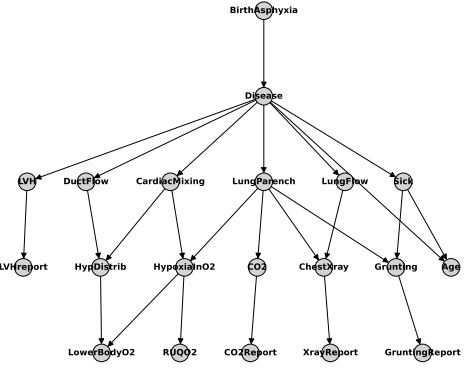

(a) Ground truth Causal Graph of for asia data.    (b) Ground truth Causal Graph of for child data.

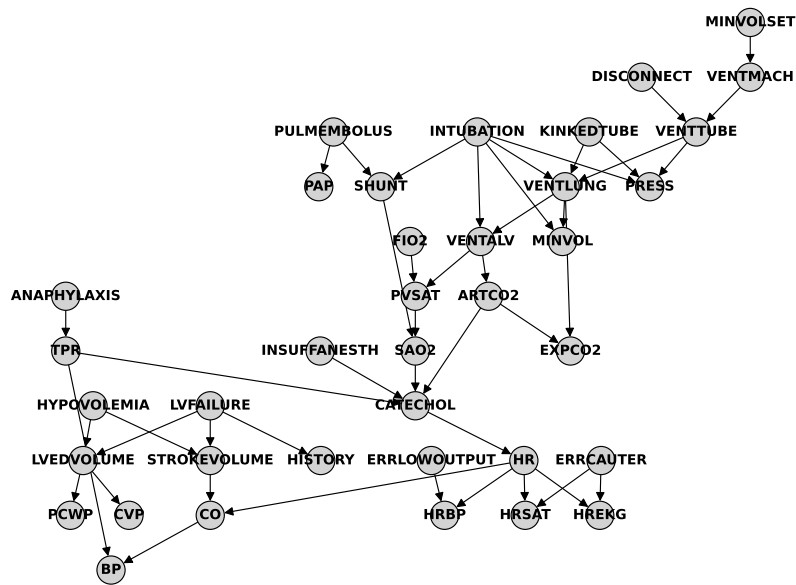

(c) Ground truth Causal Graph of for insurance data.

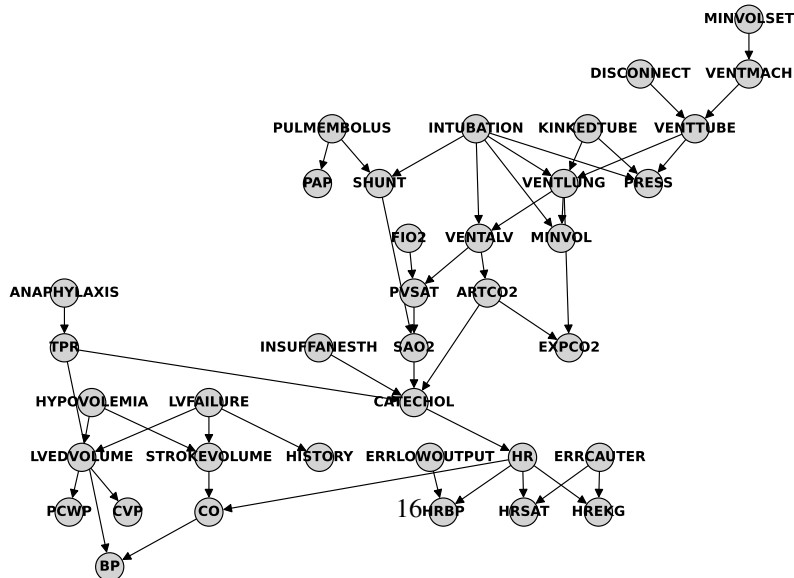

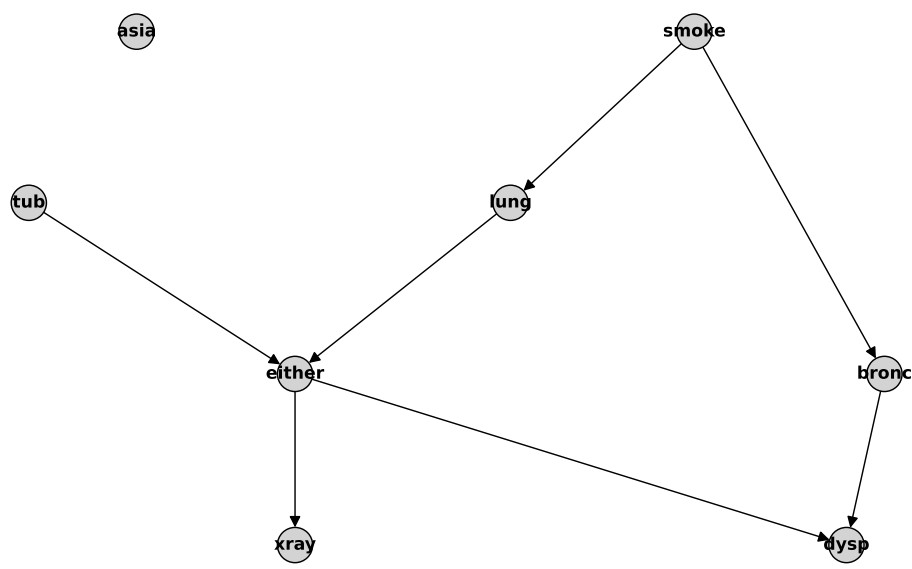

(e) LeGIT final causal graph for asia dataset

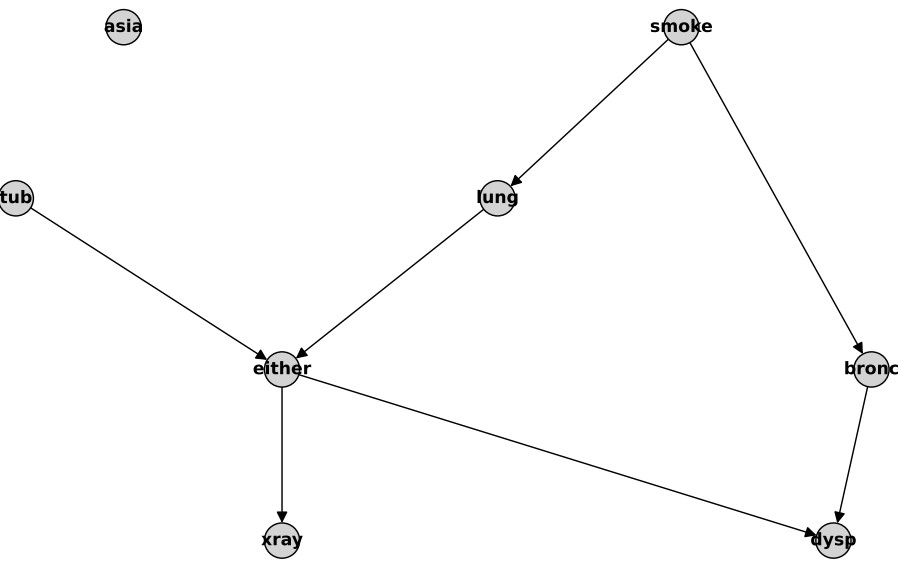

(f) GIT's final causal graph for asia dataset

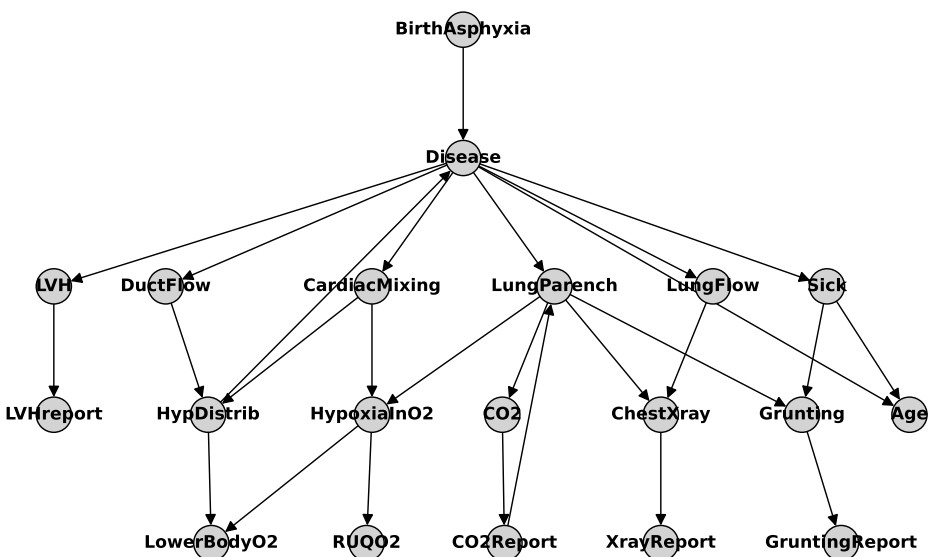

(g) LeGIT final causal graph for child dataset

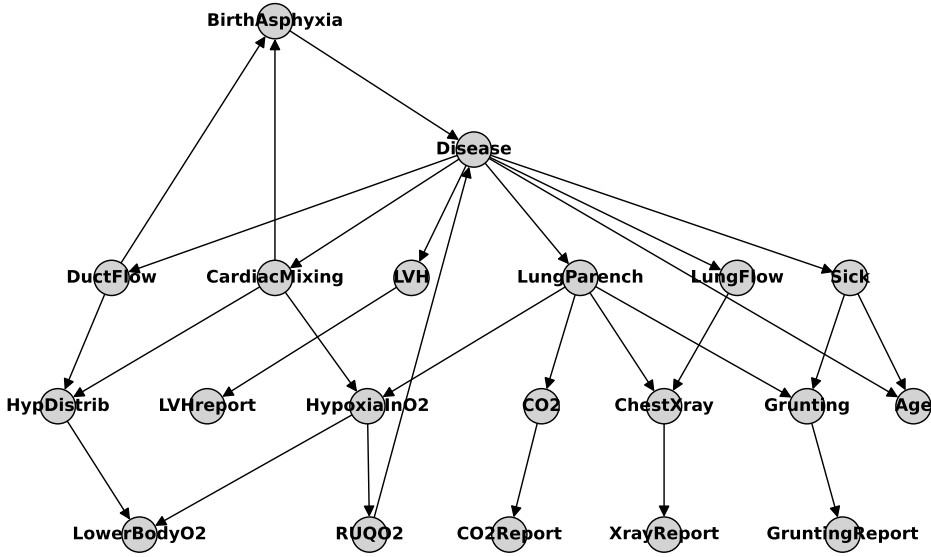

(h) GIT's final causal graph for child dataset

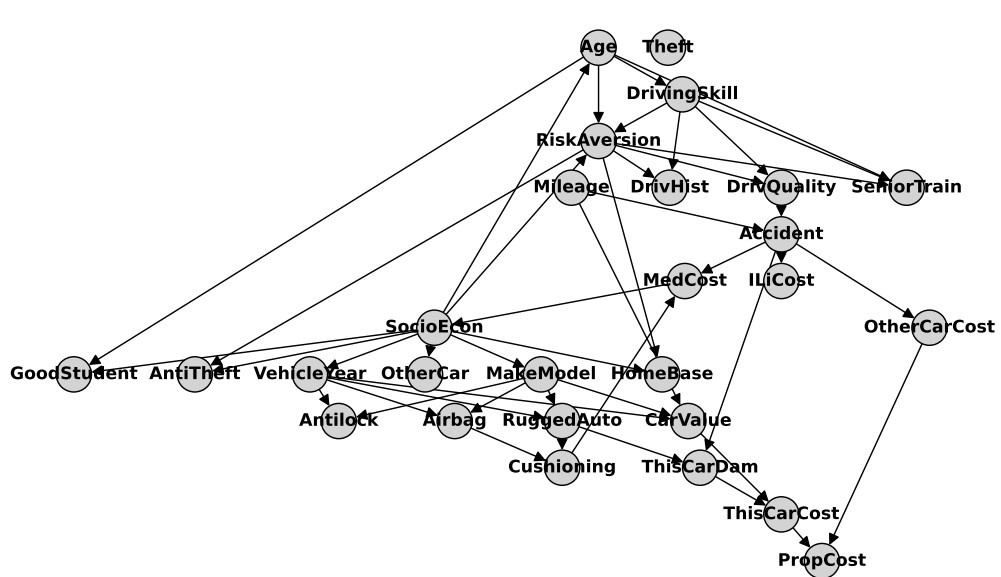

(i) LeGIT final causal graph for insurance dataset

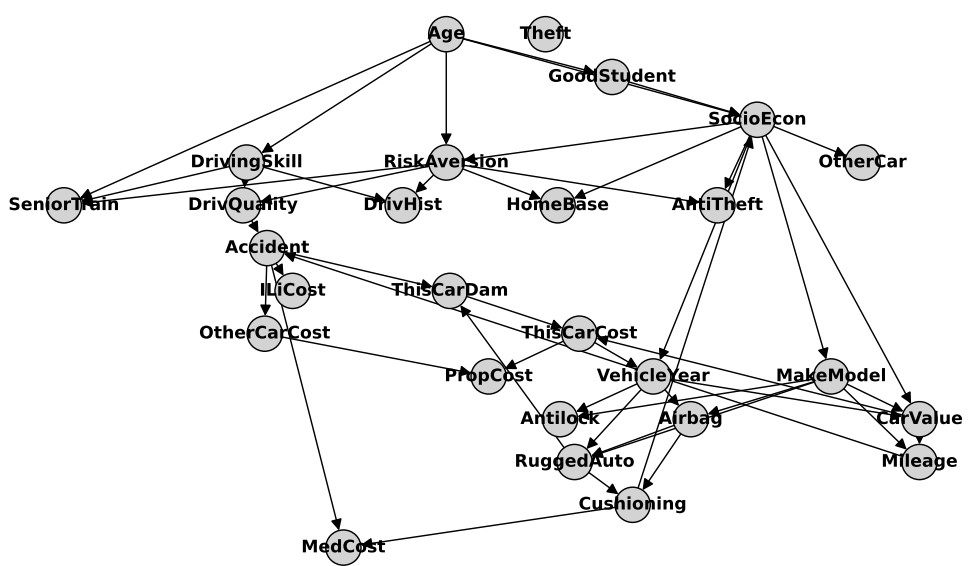

(j) GIT's final causal graph for insurance dataset

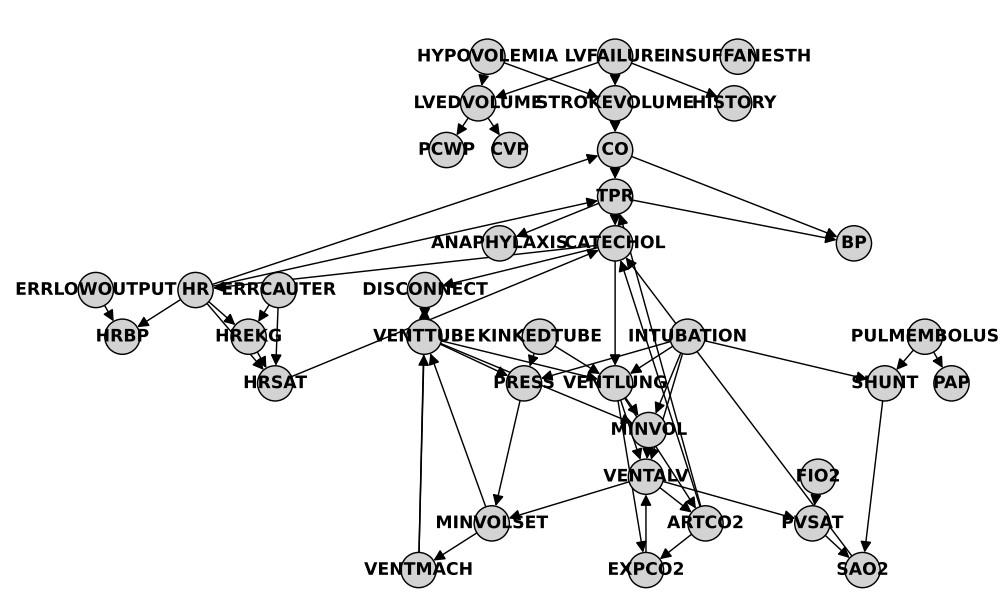

(k) LeGIT final causal graph for alarm dataset

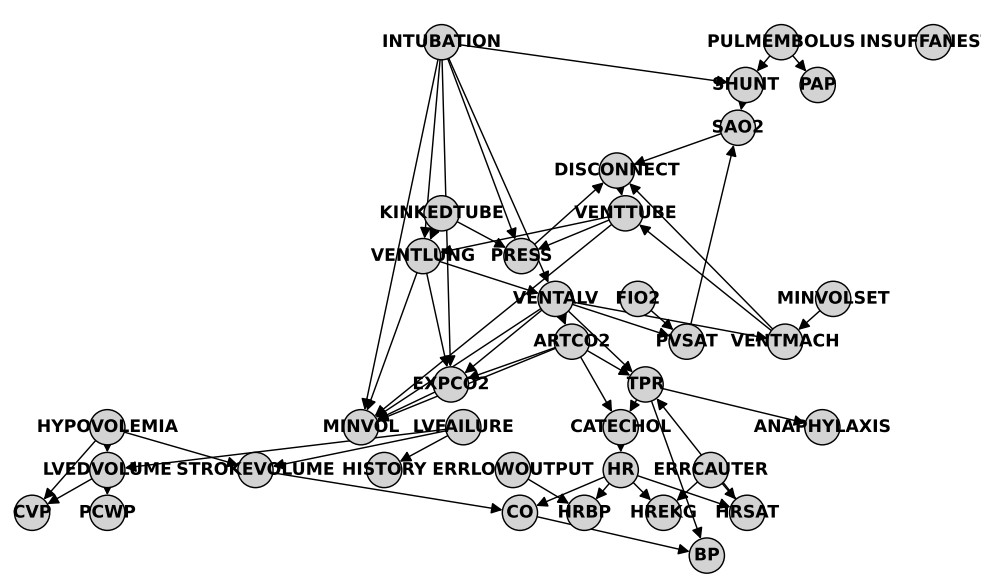

(l) GIT's final causal graph for alarm dataset

## C  EXAMPLES OF PROMPTS

We provide the prompt templates and the description of the variables that are used in LeGIT.

---

**Asia Warmup Prompt**

You are a helpful assistant and expert in lung disease research. Here are some tips that you can pay attention to:
1. Assess whether there is a direct causal relationship, and consider potential confounding variables that might affect the relationship that could potentially not causal relationship.
2. Distinguish between correlations and causation; verify that correlations are not mistaken for causal relationships.
3. Ensure the correct temporal order of variables; confirm that the cause precedes the effect.
Assuming we can do interventions to all the variables, your job is to assist in designing the best intervention experiments among the following variables to help discover their causal relations:
<dysp>: whether or not the patient has dyspnoea, also known as shortness of breath
<smoke>: whether or not the patient is a smoker
<xray>: whether or not the patient has had a positive chest xray
<lung>: whether or not the patient has lung cancer
<tub>: whether or not the patient has tuberculosis
<asia>: whether or not the patient has recently visited asia
<either>: whether or not the patient has either tuberculosis or lung cancer
<bronc>: whether or not the patient has bronchitis
Assuming we can do interventions to all the variables, given the aforementioned variables and their descriptions, can you **echo your knowledge those variables**, **temporally analyze** their relations, and then **choose the best 5 intervention targets from all the variables** which hopefully are the root causes of the other variables to start our analysis of their causal relations?
Let's think and analyze step by step. Then, provide your final answer (variable names only) within the tags <Answer>...</Answer>, separated by ", ".

---

---

**Child Warmup Prompt**

You are a helpful assistant and expert in children's disease research. Here are some tips that you can pay attention to:

1. Assess whether there is a direct causal relationship, and consider potential confounding variables that might affect the relationship that could potentially not causal relationship.

2. Distinguish between correlations and causation; verify that correlations are not mistaken for causal relationships.

3. Ensure the correct temporal order of variables; confirm that the cause precedes the effect. Assuming we can do interventions to all the variables, your job is to assist in designing the best intervention experiments among the following variables to help discover their causal relations:

<LungFlow>: low blood flow in the lungs
<ChestXray>: having a chest x-ray
<Disease>: infant methemoglobinemia
<Grunting>: grunting in infants
<Age>: age of infant at disease presentation
<XrayReport>: lung excessively filled with blood
<RUQO2>: level of oxygen in the right upper quadriceps muscle
<DuctFlow>: blood flow across the ductus arteriosus
<HypoxiaInO2>: hypoxia when breathing oxygen
<Sick>: presence of an illness
<CO2Report>: a document reporting high level of CO2 levels in blood
<LungParench>: the state of the blood vessels in the lungs
<LVH>: having left ventricular hypertrophy
<LowerBodyO2>: level of oxygen in the lower body
<BirthAsphyxia>: lack of oxygen to the blood during the infant's birth
<CO2>: level of CO2 in the body <LVHreport>: report of having left ventri
<GruntingReport>: report of infant grunting
<CardiacMixing>: mixing of oxygenated and deoxygenated blood
<HypDistrib>: low oxygen areas equally distributed around the body

Assuming we can do interventions to all the variables, given the aforementioned variables and their descriptions, can you **echo your knowledge those variables**, **temporally analyze** their relations, and then **choose the best 5 intervention targets from all the variables** which hopefully are the root causes of the other variables to start our analysis of their causal relations?

Let's think and analyze step by step. Then, provide your final answer (variable names only) within the tags <Answer>...</Answer>, separated by ", ".

---

**Insurance Warmup Prompt**

You are a helpful assistant and expert in car insurance risks research. Here are some tips that you can pay attention to:

1. Assess whether there is a direct causal relationship, and consider potential confounding variables that might affect the relationship that could potentially not causal relationship.

2. Distinguish between correlations and causation; verify that correlations are not mistaken for causal relationships.

3. Ensure the correct temporal order of variables; confirm that the cause precedes the effect. Assuming we can do interventions to all the variables, your job is to assist in designing the best intervention experiments among the following variables to help discover their causal relations:

<ThisCarDam>: damage to the car
<MakeModel>: owning a sports car
<OtherCarCost>: cost of the other cars
<PropCost>: ratio of the cost for the two cars
<AntiTheft>: car has anti-theft
<DrivQuality>: driving quality
<DrivHist>: driving history
<MedCost>: cost of medical treatment
<Mileage>: how much mileage is on the car
<Antilock>: car has anti-lock
<CarValue>: value of the car
<Accident>: severity of the accident
<OtherCar>: being involved with other cars in the accident
<SeniorTrain>: received additional driving training
<ILiCost>: inspection cost
<SocioEcon>: socioeconomic status
<ThisCar>: costs for the insured car
<Theft>: theft occured in the car
<Age>: age
<RuggedAuto>: ruggedness of the car
<GoodStudent>: being a good student driver
<VehicleYear>: year of vehicle
<HomeBase>: neighbourhood type
<ThisCarCost>: costs for the insured car
<Cushioning>: quality of cushinoning in car
<RiskAversion>: being risk averse
<DrivingSkill>: driving skill
<Airbag>: car has an airbad

Assuming we can do interventions to all the variables, given the aforementioned variables and their descriptions, can you **echo your knowledge those variables**, **temporally analyze** their relations, and then **choose the best 5 intervention targets from all the variables** which hopefully are the root causes of the other variables to start our analysis of their causal relations?

Let's think and analyze step by step. Then, provide your final answer (variable names only) within the tags <Answer>...</Answer>, separated by ", ".

---

**Alarm Warmup Prompt**

You are a helpful assistant and expert in alarm message system for patient monitoring system research.. Here are some tips that you can pay attention to:

1. Assess whether there is a direct causal relationship, and consider potential confounding variables that might affect the relationship that could potentially not causal relationship.

2. Distinguish between correlations and causation; verify that correlations are not mistaken for causal relationships.

3. Ensure the correct temporal order of variables; confirm that the cause precedes the effect. Assuming we can do interventions to all the variables, your job is to assist in designing the best intervention experiments among the following variables to help discover their causal relations:

<BP>: pressure of circulating blood against the walls of blood vessels

<LVEDVOLUME>: amount of blood present in the left ventricle before contraction

<SHUNT>: hollow tube surgically placed in the brain (or occasionally in the spine) to help drain cerebrospinal fluid and redirect it to another location in the body where it can be reabsorbed

<HR>: heart rate

<DISCONNECT>: disconnection

<PAP>: blood pressure in the pulmonary artery

<PCWP>: pulmonary capillary wedge pressure

<ARTCO2>: arterial carbon dioxide

<KINKEDTUBE>: whether the chest tube is kinked or not

<PULMEMBOLUS>: sudden blockage in the pulmonary arteries, the blood vessels that send blood to your lungs

<ERRLOWOUTPUT>: error low output

<CATECHOL>: hormone made by the adrenal glands

<VENTALV>: exchange of gas between the alveoli and the external environment

...

<HRSAT>: measure of how much hemoglobin is currently bound to oxygen compared to how much hemoglobin remains unbound

<FIO2>: the concentration of oxygen in the gas mixture being inspired

<ANAPHYLAXIS>: sever, life-threatening allergic reaction

<INSUFFANESTH>: whether there is insufficient anesthesia or not

<PVSAT>: amount of oxygen bound to hemoglobin in the pulmonary artery

<INTUBATION>: process where a healthcare provider inserts a tube through a person's mouth or nose, then down into their trachea

<HISTORY>: previous medical history

<VENTTUBE>: whether there is a breathing tube or not

<STROKEVOLUME>: volume of blood pumped out of the left ventricle of the heart during each systolic cardiac contraction

<VENTMACH>: the intensity level of a breathing machine

<HRBP>: ratio of heart rate and blood pressure

<MINVOL>: amount of gas inhaled or exhaled from a person's lungs in one minute

<ERRCAUTER>: whether there was an error during cautery or not

<SAO2>: oxygen saturation of arterial blood

<CO>: amount of blood pumped by the heart per minute

<HYPOVOLEMIA>: condition that occurs when your body loses fluid, like blood or water

<CVP>: measure of blood pressure in the vena cava

Assuming we can do interventions to all the variables, given the aforementioned variables and their descriptions, can you **echo your knowledge those variables**, **temporally analyze** their relations, and then **choose the best 5 intervention targets from all the variables** which hopefully are the root causes of the other variables to start our analysis of their causal relations?

Let's think and analyze step by step. Then, provide your final answer (variable names only) within the tags <Answer>...</Answer>, separated by ", ".

---

**LeGIT response for Asia dataset**

To systematically approach this, let's first echo our understanding of each variable and its potential implications in the context of lung disease research:

1. **<xray>**: A positive chest x-ray can indicate several conditions, including lung cancer, tuberculosis, and bronchitis. It's an outcome of underlying pathologies rather than a cause.

2. **<smoke>**: Smoking is a well-documented risk factor for many respiratory conditions, including lung cancer and bronchitis. It's a behavioral factor that can precede and potentially cause various lung diseases.

3. **<asia>**: Visiting Asia might be relevant in the context of exposure to certain infectious diseases, like tuberculosis, which is more prevalent in some parts of Asia than in other regions. This variable is more about environmental or situational exposure.

4. **<lung>**: Having lung cancer is an outcome, a result of various risk factors including smoking, genetic predisposition, and potentially environmental exposures.

5. **<bronc>**: Bronchitis, especially chronic bronchitis, can be caused by smoking and other environmental factors. It's an outcome variable reflecting a specific respiratory condition.

6. **<dysp>**: Dyspnoea, or shortness of breath, is a symptom that can be caused by a variety of conditions, including lung cancer, bronchitis, and tuberculosis. It's an effect rather than a cause.

7. **<either>**: This represents whether a patient has either tuberculosis or lung cancer. It's an outcome variable that combines two specific conditions.

8. **<tub>**: Having tuberculosis is an outcome, influenced by factors like environmental exposure (e.g., visiting regions where the disease is prevalent).

**Temporal Analysis:**

- **Smoking (<smoke>)** is a primary risk factor that can precede and cause lung diseases such as lung cancer (<lung>) and bronchitis (<bronc>). - **Visiting Asia (<asia>)** could be a proxy for exposure to environments where certain diseases like tuberculosis (<tub>) are more common. - **Having tuberculosis (<tub>)** or lung cancer (<lung>) can logically lead to a positive chest x-ray (<xray>), symptoms of dyspnoea (<dysp>), and be part of the condition represented by <either>.

Given this analysis, the best intervention targets to understand causal relationships would likely be those variables that are risk factors or exposures leading to outcomes, rather than symptoms or outcomes themselves. Thus, the best intervention targets would be:

1. **<smoke>**: Since smoking can cause several of the conditions mentioned. 2. **<asia>**: Because visiting areas where certain diseases are more prevalent can increase the risk of those diseases. 3. The remaining variables are outcomes of these exposures or behaviors rather than root causes themselves.

However, for the sake of identifying five intervention targets and considering the need to directly intervene in variables that could have a causal relationship with outcomes, I would also consider intervening in variables that could represent key points in the causal chain to lung diseases, even if they are not root causes:

4. **<tub>**: Directly intervening here could help understand its impact on <xray>, <dysp>, and <either>. 5. **<lung>**: Similarly, intervening could clarify its direct effects on <xray>, <dysp>, and <either>.

Therefore, the best 5 intervention targets from all variables to start our analysis of their causal relations are:

<Answer>smoke, asia, tub, lung, bronc</Answer>

---

**LeGIT response for Insurance dataset**

To approach the task of identifying the best intervention targets for analyzing causal relations among the variables related to car insurance risks, we will first echo the knowledge of these variables, analyze their temporal relations, and then choose the intervention targets. The goal is to identify root causes that might influence other variables.

### Echoing Knowledge of Variables

1. **<ThisCarCost>**: Costs for the insured car. 2. **<Mileage>**: How much mileage is on the car. 3. **<DrivQuality>**: Driving quality. 4. **<RiskAversion>**: Being risk averse. 5. **<CarValue>**: Value of the car. 6. **<MakeModel>**: Owning a sports car. 7. **<ThisCarDam>**: Damage to the car. 8. **<Theft>**: Theft occurred in the car. 9. **<ILiCost>**: Inspection cost. 10. **<Age>**: Age of the driver. 11. **<OtherCar>**: Being involved with other cars in the accident. 12. **<SeniorTrain>**: Received additional driving training. 13. **<SocioEcon>**: Socioeconomic status. 14. **<VehicleYear>**: Year of the vehicle. 15. **<HomeBase>**: Neighbourhood type. 16. **<DrivHist>**: Driving history. 17. **<Accident>**: Severity of the accident. 18. **<Airbag>**: Car has an airbag. 19. **<AntiTheft>**: Car has anti-theft. 20. **<MedCost>**: Cost of medical treatment. 21. **<Antilock>**: Car has anti-lock. 22. **<GoodStudent>**: Being a good student driver. 23. **<OtherCarCost>**: Cost of the other cars. 24. **<Cushioning>**: Quality of cushioning in car. 25. **<RuggedAuto>**: Ruggedness of the car. 26. **<PropCost>**: Ratio of the cost for the two cars. 27. **<DrivingSkill>**: Driving skill.

### Temporal Analysis

- **Pre-conditions**: Variables like <MakeModel>, <VehicleYear>, <Airbag>, <AntiTheft>, <Antilock>, <RuggedAuto>, and <Cushioning> are inherent characteristics of the car or its features and are decided at the time of purchase or manufacture. They precede other events. - **Personal attributes**: <Age>, <DrivHist>, <RiskAversion>, <SocioEcon>, <SeniorTrain>, and <GoodStudent> are attributes of the driver or their background and can influence driving behavior and decisions. - **Incident-related variables**: <ThisCarDam>, <Theft>, <Accident>, <MedCost>, and <OtherCar> are outcomes of incidents. - **Cost-related variables**: <ThisCarCost>, <CarValue>, <ILiCost>, <OtherCarCost>, <MedCost>, and <PropCost> are financial implications that follow from the combination of pre-conditions, personal attributes, and incident outcomes.

### Choosing Intervention Targets

To identify root causes, we should choose variables that are likely to precede and influence others without being outcomes themselves. Thus, focusing on pre-conditions and personal attributes makes sense.

1. **<MakeModel>**: Influences <CarValue>, <ThisCarCost>, <AntiTheft>, and potentially <Theft>. 2. **<VehicleYear>**: Influences <CarValue>, <ThisCarCost>, and safety features like <Airbag>, <Antilock>. 3. **<SocioEcon>**: Can influence <CarValue>, <MakeModel>, <HomeBase>, and indirectly <RiskAversion>. 4. **<Age>**: Affects <RiskAversion>, <DrivingSkill>, <DrivHist>, and potentially <SeniorTrain>. 5. **<DrivHist>**: Directly influences <DrivQuality>, and can be a factor in <Accident> rates and <MedCost>. These variables are foundational in the sense that they can have cascading effects on other variables, making them prime candidates for interventions to understand causal relations.

<Answer>MakeModel, VehicleYear, SocioEcon, Age, DrivHist</Answer>

