# OpenReview forum: "Can Large Language Models Help Experimental Design for Causal Discovery?"
_ICLR.cc/2025/Conference — Submitted to ICLR 2025_

### Official Review · Reviewer_graM · 2024-10-21

**Soundness:** 2
**Presentation:** 2
**Contribution:** 2
**Rating:** 5
**Confidence:** 3

**Summary:**

This work investigates whether the world knowledge possessed by large language models (LLMs) can assist in identifying better intervention targets for causal discovery tasks. The author provides an algorithm for this task and empirically shows that, with the assistance of LLMs, it is indeed able to identify better causal structures in fewer iterations compared to existing methods.

**Strengths:**

1. Positive empirical results: This work provides a simple and useful algorithm for identifying better causal structures.
2. Clarity: This work provides a simple technique, via using LLM for a warm start for choosing intervention targets, and thus the work is easy to follow.

**Weaknesses:**

1. Clarification Needed on Section 4.1: I find the arguments in Section 4.1 unclear. The authors suggest that selecting nodes with more edges is more beneficial compared to nodes with fewer edges. However, based on Figure 2, it seems that the proposed method, LeGit, did not consistently select nodes with more edges, particularly in the child dataset.

2. Concerns about Table 2: While Table 2 shows that LeGit achieves lower SHD scores, the standard deviation is quite high, which I feel less confidence in this method’s effectiveness in low-data scenarios.

3. Request for BIC Scores: BIC is a standard metric for assessing the quality of a discovered causal graph. Could you also provide BIC scores as an additional metric?

**Questions:**

See weaknesses session.

**Details Of Ethics Concerns:**

I believe this work does not have any ethical concerns.

---

> ### Comment · Reviewer_graM · 2024-11-25
>
> Dear Authors,
>
> Thank you so much for your response and clarification. Once again, as per Q1, by looking at section 4.1 and **the corresponding Figure 2**, it appears that the claim, as depicted in section 4.1, that selecting nodes with more edges is more beneficial compared to nodes with fewer edges and **LeGit** can identify such nodes **is not aligned with what is exactly shown in Figure 2**. In addition, if the benefits of **LeGit** in identifying influential nodes only appear in later epochs, it might be better to show results in later iterations to support your claim than the current plot.
>
> Perhaps, also by looking at other reviews, the authors would need major revisions to improve this work.
>
> Therefore, I keep my current score.
>
> Once again, I appreciate the authors' detailed response and clarification.
>
> Bests,
>
> Reviewer graM

---

### Official Review · Reviewer_Bo1N · 2024-10-27

**Soundness:** 3
**Presentation:** 3
**Contribution:** 2
**Rating:** 3
**Confidence:** 3

**Summary:**

The submission builds on previous work actively selecting intervention nodes for causal dscovery, proposing an initial warmup phase where an LLM selects intervention nodes before the numerical method itself chooses the nodes.

**Strengths:**

Originalty : it is natural to consider LLMs to choose intervention nodes but it is indeed the first paper I see doing it.

Quality : the method is explained in thorough detail, as well as its motivation.

Clarity : the paper is mostly clear in how the method is presented.

**Weaknesses:**

Clarity : the plots with node frequency are slightly hard to parse, especially as the reader has to manually count all their neighbors. Further some names are given without locating them in the graph in the main part (eg SocioEcon).

Signifiance : the improvement of the warmup phase wrt GIT seems rather weak, as the SHDs in Tables 1 and 2 are almost always within standard deviations. Also, the ablation study seems to only discuss the Insurance dataset, I do not see any further comment on the other datasets. And actually I am not sure to comprehend how this is an ablation study! Indeed, Tables 1 and 2  in Section 5.2 already involve a form of ablation study, as GIT can be thought as the method of submission where the LLM part has been ablated.

Quality : the theoretical analysis is contained in a few lines, and seems to mostly consist in justifying convergence of the method thanks to the convergence of the original numerical methods. This section would be significantly improved if a theoretical proof of whether / how much the LLM-selected nodes make convergence faster, e.g. a quantification of the error rate based on the number oif LLM-selected nodes.

**Questions:**

- Figure 2 : it is claimed that for GIT, "the selected node simply influences few nodes" (l.257-258) and "It can be found that, given only the meta information, LLMs are able to relate the rich world knowledge to locate the desired influential nodes." (l.268-269). Can you explain why these specific nodes are the most desired and influential? It looks like neighbors of GIT's highly selected node (red) also are neighbors with many nodes. Even better, could you explicitly provide a metric for desirability? (even something as simple as the number of neighbors)

- Figures 1, 6&7 : could you add numbers of neighbors of every node, either in the plot directly or in a separate table?

- How do Figures 6 and 7 justify that numerical-only methods "get trapped in the initialization phase of the Insurance dataset"? From my understanding, the distributions of node frequencies are very similar between LeGIT and its numerical competitors, as all nodes except one for AIT receive blue assignments.
 Tied to an above question, could you explicitly quantify the desirability of the nodes selected by LeGIT and the baselines, highlighting that the former improves over the latter?

- Figures 6&7 : what comments would you have on other datasets? (or explain why you only comment on the Insurance dataset?)

- Section 5.3 : as an ablation study, could you also consider using only the LLM warmup and not the numerical selection of interventions?

---

### Official Review · Reviewer_vDEa · 2024-10-28

**Soundness:** 2
**Presentation:** 2
**Contribution:** 2
**Rating:** 5
**Confidence:** 2

**Summary:**

The authors leverage Large Language Models (LLMs) to assist with the intervention targeting in causal discovery by making use of the rich world knowledge about the experimental design in LLM. They present Large Language Model Guided Intervention Targeting (LeGIT), a framework that effectively incorporates LLMs to assist with the intervention targeting in causal discovery.

**Strengths:**

Across 4 different scales of realistic benchmarks, LeGIT significantly outperforms previous approaches.

**Weaknesses:**

Please refer to the questions

**Questions:**

While the authors claim that LeGIT opens up a new frontier for using LLMs in experimental design, there is no theoretical guarantee.

It seems that the convergence of numerical-based methods does not imply convergence of LeGIT. In addition, what is the convergence limit of the algorithm?

---

### Official Review · Reviewer_iBNB · 2024-11-05

**Soundness:** 2
**Presentation:** 3
**Contribution:** 2
**Rating:** 3
**Confidence:** 3

**Summary:**

In many settings, causal structure learning from observational data alone isn't possible.  In settings where we can intervene, interventions must still be chosen carefully, since they may be expensive to perform.  There are existing methods that propose intervention targets for an online learning setting, but suffer from limitations in practice.  The authors propose an algorithm that uses an LLM to select intervention targets.  By providing the LLM with information about the domain and variables, lists of potential root causes can be obtained and used as intervention targets.  The authors show that the LLM is often able to select better intervention candidates than GIT, and they show empirically that LeGIT is competitive with GIT on multiple baseline empirical datasets.

**Strengths:**

The paper is generally well-written and easy to follow.  The terminology is clear, and I think the inclusion of the ENCO algorithm block sets up the online learning setting very well.  Algorithm 2 is easily understandable, and the results tables and figures are clear and well-presented.  The authors present the online learning problem well in the introduction, and present a nice discussion around the challenges in the setting, as well as the potential benefits and challenges posed by the incorporation of LLMs.

**Weaknesses:**

My biggest concern with this paper is that I'm struggling to see the use case and what value the LLM is really providing here.  The LLM appears to be acting in the role of a human expert, suggesting which nodes are likely to be the highest up in the network based on their semantic meaning.  At first glance, the benefit of the LLM is that it's readily available and the process could be automated.  However, online causal structure learning in an intervenable system isn't generally the type of process that is fully automated by people with little domain knowledge, so some (though certainly not all) of that value is diminished.

This system also seems to require that the system being analyzed, and all of the measured variables, fall somewhat into "common knowledge", or are at least discussed enough in the LLM's training data to enable it to make reasonable inferences.  Even if the particular domain is discussed in the LLM's training data, any given domain may have nuance to its features or data collection mechanism that are unique and may be hard to communicate to an LLM.  This isn't necessarily a dealbreaker, but for these challenges to be worth it, the LLM needs to be providing a substantial value beyond what a human could, and I'm not sure I'm seeing that value at this point.

While human experts are not particularly scalable, the fact that LeGIT is designed to work with a system in which a large portion of the variables can be directly intervened on suggests that, in any situation where LeGIT would be deployed, there would be humans on hand who are at least familiar enough with the system to set up that intervention apparatus.  What's missing for me from this paper, then, is a comparison with human-suggested intervention targeting.  Since the datasets considered in this paper are fairly human understandable, I don't think you'd even need someone who is particularly an "expert".  I'd be curious to know how LeGIT performance compares to a run where, whenever the LLM would be queried, it instead presents the variable names and descriptions to a reasonably-informed human to propose potential root causes.

Some other, more specific, notes:

On line 88, the authors point to Figure 1 as showing that numerical methods start with limited information about the underlying system.  However, I'm not sure where I'm supposed to see that in Figure 1.  Figure 1 as a whole is also just not very informative.  Part (a), Warmup intervention, makes sense.  But then parts (b) and (c) are "Intervention" and "Continual intervention", with nearly the same picture.  The caption seems to suggest that steps a and b are repeated while step c is not ("through multiple rounds of steps (a) and (b)"), but nothing in the picture suggests that, and term "continual intervention" seems to imply a looped process more than just "intervention".

I think Figure 1 is perhaps trying to show that an LLM (a) is used to select a single intervention (b), which is used as part of a continual learning process.  However, that is not clear at all from the figure, which seems to show a single pass system, and makes it look like a single LLM-selected intervention (b) looks basically exactly the same as...the whole continual learning process?  Essentially, Figure 1 needs to be completely reworked.

In lines 198-199, you mention the hard vs soft intervention distinction.  Weirdly, though, you never refer to hard or soft interventions again, not even to specify which one you use in your experiments.  Also, if you are going to define hard vs soft interventions, you should actually defined soft interventions.  Just saying "Otherwise soft" is not sufficient at all.

I found the results in 4.1 somewhat interesting but ultimately not that convincing.  The comparison does seem to suggest that LeGIT is able to find more upstream nodes than GIT.  However, since you're only comparing against a single method for intervention selection, and just showing three images, it's hard to know how general this finding is, both to other graphs and, more importantly, to other intervention targeting methods.  Are there other intervention targets you can compare against here, to see how well they do at selecting upstream nodes in the network?

The authors claim that, according to Figure 4, despite not performing as well with lower data sample sizes, "LeGIT converges to a better solution faster than other methods".  I cannot figure out where this statement comes from.  It looks to be, at best, competitive with GIT.  For child, maybe you can say that it outperforms at the higher data sizes.  However, at the end, it's only an SHD difference of ~1, so they look essentially equivalent.  For alarm, GIT is better for all but the final point in the graph, and for insurance, the two appear competitive.  Are there other results that you're basing your statement there off of?  If I'm understanding correctly, it looks likeTable 1 is showing the final point in Figure 4, and looking at the standard deviations, GIT and LeGIT are well within 1 standard deviation of each other for all datasets.

The authors say multiple times that inferring causal structures from observational data alone is impossible.  However, I thought the general idea was that it's impossible *without additional assumptions*, not just fully impossible.

**Questions:**

Can you provide an example of what the variable descriptions fed to the LLM look like?

---

### Official Review · Reviewer_SDxB · 2024-11-05

**Soundness:** 3
**Presentation:** 3
**Contribution:** 2
**Rating:** 3
**Confidence:** 4

**Summary:**

The paper proposes and evaluates a “targeting method” — a method to identify experimental interventions that will be maximally informative in the construction of a causal graphical model. The approach (LeGIT) builds on an existing method (GIT) by incorporating the use of a large language model.

**Strengths:**

The general problem is well-motivated and the approach is well described within the text of the paper.

The inclusion of learning curves (Figure 4) is a very good idea. It provides a far richer picture of the performance of LeGIT than the more traditional table of results (e.g., Table 1).

**Weaknesses:**

The statement of research question (“Can we incorporate the knowledge of LLMs to assist with intervention targeting?”) is unhelpful. Whether something is possible is of less interest than how and why it works. A better phrasing would be something like “Under what conditions does LLM-based targeting of interventions improve causal inference?”.

The evaluation results indicate that GIT and LeGIT are not meaningfully different. First, the point estimates of model accuracy (SHD) for the two systems provided in Table 1 are well within the confidence intervals of each other. Second, it is clear that GIT attains low SHD much more quickly than LeGIT. If you had less data, then GIT would fairly clearly be the better method to use. The paper claims that “…despite a faster decrease speed of GIT, GIT finally converges to a suboptimal solution…” That may be true, but there is no evidence that LeGIT converges to any better solution. Furthermore, the comparison between GIT and LeGIT is particularly important, given that, as I understand it, (roughly) LeGIT = GIT + LLM. If the addition of an LLM provides no substantial advantage (and increases error at smaller sample sizes), then there seems to be no reason to prefer LeGIT.

Even if LeGIT really does reduce error, one way of summarizing those results is “prior knowledge is useful when selecting experiments”. That’s not unexpected. The key question is whether current LLMs provide a useful set of such prior knowledge that would not already be evident to human analysts. I don’t know the answer, but the work reported in the current paper doesn’t improve our knowledge about this question.

SHD has some known issues that make it a less useful evaluation metric than alternatives, including the structural intervention distance (Peters & Bühlmann 2015) and the balanced scoring function (Constantinou 2019). The authors should consider comparing their results when using those metrics.

The “Related Work” section describes several approaches for selecting intervention targets, including AIT, GIT, CBED, and CALOID. However, only AIT and GIT are included as baselines in the evaluation.

The idea of conducting ablation studies (Section 5.3) is great. However, it is entirely unclear how the results reported in this section are related to an ablation study which would, presumably, remove or replace (“ablate”) certain portions of the system.

The authors should examine what happens if some assumptions are not met. For example, what happens if common causal notions (encoded in LLMs) are incorrect? What effect does that have on the output of the system?

References

Peters, J., & Bühlmann, P. (2015). Structural intervention distance for evaluating causal graphs. *Neural computation*, *27*(3), 771-799.

Constantinou, A. C. (2019). Evaluating structure learning algorithms with a balanced scoring function. *arXiv preprint arXiv:1905.12666*.

**Questions:**

What prevents LeGIT from getting to zero SHD? In Figure 4, for the Alarm and Insurance domains, it appears that LeGIT has asymptoted at a non-zero SHD. Why does this happen, given that experiments should be able to resolve all uncertainty in model structure?

Why aren’t the results for “Asia” included in Figure 4 and Table 2?

Why weren’t CBED and CALOID included as baselines?

---

### Meta-Review · Area_Chair_ejSR · 2024-12-21

**Metareview:**

The paper proposes a method that uses Large Language Models to guide intervention targeting in causal discovery.


Strengths:

+ Studies important problem of how to leverage LLMs for selecting intervention nodes in causal discovery

Weaknesses:

+ Lacks significant empirical improvement; results show LeGIT is not meaningfully better than GIT, with differences within standard deviation

+ Unclear if the proposal is meaningful since the LLM appears to just replicate what a human expert could do, without clear advantages

+ Lack of theoretical guarantees and analysis, especially regarding convergence properties and performance bounds

**Additional Comments On Reviewer Discussion:**

The reviewers are in agreement that the paper in its current form does not meet the threshold for acceptance due to limited demonstrated improvements and the lack of theoretical foundations.

---

### Decision · Program_Chairs · 2025-01-22

Reject